# CK2 Chemical Probes: Past, Present, and Future

Han Wee Ong [1], David H. Drewry [1,2] and Alison D. Axtman [1,*]

1   Structural Genomics Consortium, UNC Eshelman School of Pharmacy, University of North Carolina at Chapel Hill, Chapel Hill, NC 27599, USA; onghw@live.unc.edu (H.W.O.); david.drewry@unc.edu (D.H.D.)
2   UNC Lineberger Comprehensive Cancer Center, UNC School of Medicine, University of North Carolina at Chapel Hill, Chapel Hill, NC 27599, USA
*   Correspondence: alison.axtman@unc.edu

**Abstract:** Protein kinase casein kinase 2 (CK2/CSNK2) is a pleiotropic kinase involved in many cellular processes and, accordingly, has been identified as a potential target for therapeutic intervention for multiple indications. Significant research effort has been invested into identifying CK2 inhibitors as potential drug candidates and potent and selective CK2 chemical probes to interrogate CK2 function. Here, we review the small molecule inhibitors reported for CK2 and discuss various orthosteric, allosteric, and bivalent inhibitors of CK2. We focus on the pyrazolo[1,5-*a*]pyrimidines and naphthyridines, two chemotypes that have been extensively explored for chemical probe development. We highlight the uptake and demonstrated utility of the pyrazolo[1,5-*a*]pyrimidine chemical probe **SGC-CK2-1** by the scientific community in cellular studies. Finally, we propose criteria for an ideal in vivo chemical probe for investigating CK2 function in a living organism. While no compound currently meets these metrics, we discuss ongoing and future directions in the development of in vivo chemical probes for CK2.

**Keywords:** protein kinase CK2; CSNK2A1; CSNK2A2; chemical probe; pyrazolopyrimidine; naphthyridine; orthosteric; allosteric; bivalent





## 1. Introduction

*1.1. CK2 General Structure and Small Molecule Binding Sites*

Protein kinase casein kinase 2 (CK2/CSNK2) is a constitutively active, ubiquitously expressed Ser/Thr kinase with hundreds of reported substrates [1,2]. Accordingly, CK2 plays regulatory roles in most cellular processes, and it has been identified as a potential therapeutic target for diverse diseases, including cancer, viral infections, neurodegenerative disorders, and many others [3–10]. Structurally, CK2 is a tetrameric holoenzyme composed of two catalytic subunits ($\alpha$ and $\alpha'$) and two regulatory subunits ($\beta$, Figure 1). The catalytic and regulatory subunits can assemble to make different CK2 holoenzyme combinations: $\alpha_2\beta_2$, $\alpha'_2\beta_2$, and $\alpha\alpha'\beta_2$. CK2$\alpha$ (CSNK2A1) and CK2$\alpha'$ (CSNK2A2) share high sequence similarity within their catalytic domains. However, due to differences within their C-terminal sequences, the two isoforms demonstrate some differences in their partner proteins and thus exert selected isoform-specific functions [11].

While the dimeric CK2$\beta$ regulatory subunits reinforce the stability of the intact holoenzyme, drive selection of substrates, and influence CK2 localization, there is no small molecule binding site in this domain [12–14]. Both CK2$\alpha$ and CK2$\alpha'$ have an ATP-binding site that accommodates small molecule ligands. Due to their extensive sequence identity, this orthosteric site on CK2$\alpha$ and CK2$\alpha'$ binds the same ligands with similar affinity. Many different scaffolds have been exemplified as ATP-competitive CK2 inhibitors, though they vary in potency for CK2 and kinome-wide selectivity [14–20]. Through methods such as X-ray crystallography fragment screening and virtual screening, allosteric sites on CK2 capable of binding a small molecule have been identified [14,21]. These sites include the CK2$\alpha$/CK2$\beta$ interface [21–24] and the $\alpha$D pocket [25–28], which is located near the

αD helix of the C-terminal domain of CK2α and CK2α' [14]. The development of bivalent compounds capable of simultaneously binding the CK2 ATP-binding pocket and αD pocket has been a major focus for several groups [25–28]. In addition to small molecules, peptides have also been exemplified that bind to the CK2 substrate channel as well as at the CK2α/CK2β interface [14,29,30]. Finally, bi-specific compounds that simultaneously bind to the CK2α ATP site and CK2α substrate-binding channel have been reported, but these compounds have a peptidic component and thus suffer from physicochemical limitations [28,31–34]. The various binding sites and positions within those pockets of example orthosteric, allosteric, and bivalent small molecules have been highlighted in Figure 1, with corresponding PDB codes included.

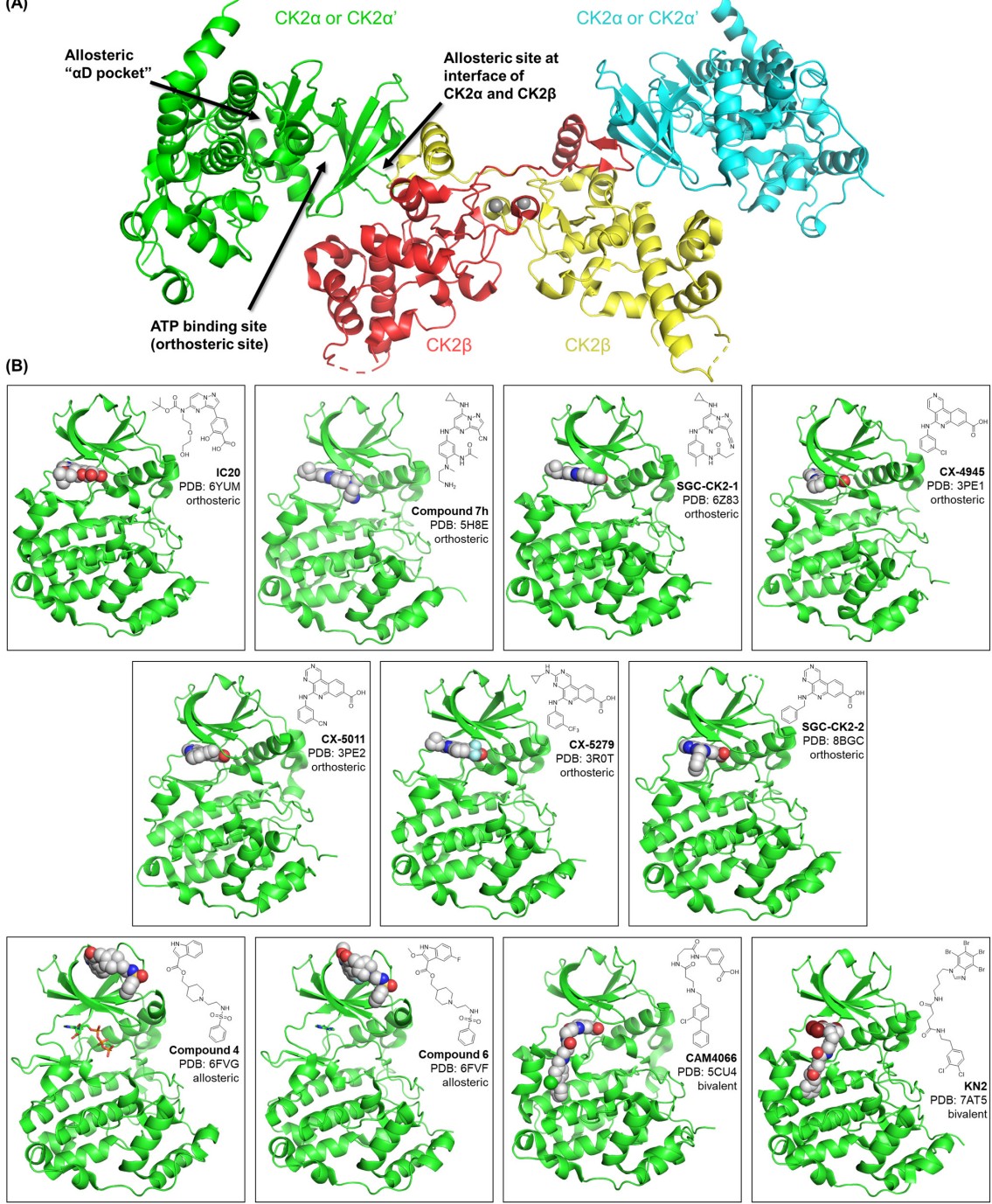

**Figure 1.** (**A**) Cartoon representation of human CK2 holoenzyme (PDB: 4MD7) with approximate positions of small molecule binding sites labeled. The holoenzyme is crystallized with two units of

CK2α (green and cyan) and two units of CK2β (red and yellow). CK2α may be substituted by CK2α' in physiological conditions. $Zn^{2+}$ ions are shown as grey spheres. (**B**) Binding sites of small molecule inhibitors (in spheres, colored by elements with carbon as white, oxygen as red, nitrogen as blue, fluorine as cyan, chlorine as green, and bromine as brown) to CK2α (green ribbon). ATP is present in the structures of compound **4** and compound **6**, shown in sticks, colored by elements with carbon as green. The binding mode and PDB code corresponding to the crystal structures are provided for each inhibitor.

### 1.2. Chemical Probe Definition

Recent emphasis has been placed on the development of inhibitors that interact selectively with CK2. This aligns with the notion that a potent and selective molecule that targets a single protein is a high-quality tool for interrogating the biological pathways regulated by that protein. A chemical probe fits these criteria and has been defined by the Structural Genomics Consortium (SGC) as meeting the metrics included in Table 1. To be approved as an SGC chemical probe, a kinase inhibitor must demonstrate a potency of <100 nM in a biochemical or biophysical assay (Table 1, #1). Enzymatic, surface plasmon resonance (SPR), isothermal calorimetry (ITC), and active site-directed competition binding assays are examples of methodologies that can and have been used to determine the cell-free potency of a putative chemical probe. Kinome-wide selectivity must be ascertained using a large human kinase panel, preferably comprising more than half of the screenable kinome (>200 kinases, Table 1, #2) [35]. The SGC kinase chemical probe selectivity goal requires that a compound demonstrates high affinity binding (percent of control <10) to <4% of the kinases within the panel when screening at a single concentration, typically 1 µM or greater [35]. Kinase chemical probes have historically exhibited better kinome-wide selectivity scores than those listed in Table 1. A deeper examination of selectivity is required, and any kinases that are inhibited and/or bind within 30-fold of the biochemical $IC_{50}/K_D$ of the primary target are considered additional targets of the chemical probe (Table 1, #3). Often near neighbor kinases, including kinases within the same kinase subfamily, fall into this category, such as AP2-associated kinase 1 (AAK1) and BMP-2-inducible kinase (BMP2K, also known as BIKE) for **SGC-AAK1-1** [36]. A kinase chemical probe typically inhibits ≤5 kinases.

Many proteins bind ATP and thus profiling against at least one other ATP-binding protein class is expected for kinase chemical probes (Table 1, #4). These compounds are sent for profiling in the PDSP GPCR panel of ~46 primary binding assays, and secondary follow-up assays are run in dose–response, where appropriate [37–39]. These data, especially for potent binders, are made available to the community as part of the data package for SGC chemical probes. For SGC kinase chemical probes, demonstration of potent cellular target engagement is required (Table 1, #5). The NanoBRET assay is typically employed to determine a cellular target engagement $IC_{50}$ value in HEK293 cells [40]. Finally, once a compound has met criteria 1–5 in Table 1, a suitable negative control is identified. This negative control must be structurally similar to the chemical probe but lack biochemical and cellular activity on the protein target(s) inhibited by the chemical probe. The use of a negative control compound alongside the chemical probe in experiments allows for ruling out chemotype-specific effects and results in the most robust science.

**Table 1.** Kinase chemical probe metrics standardized by the Structural Genomics Consortium (SGC) [41].

| SGC Kinase Chemical Probe Criteria |
| --- |
| 1. Inhibitor/agonist cell-free potency: <100 nM $IC_{50}$ or $K_D$; |
| 2. Excellent kinome-wide selectivity: $S_{10}$ (1 µM) < 0.04; |
| 3. Selectivity within kinase family: >30-fold excluding near neighbor kinases; |
| 4. Selectivity outside kinase family: describe the off-target proteins; |
| 5. On target activity in cells: <1 µM $IC_{50}$ or $EC_{50}$; |
| 6. Negative control: in vitro potency 100× less; cellular activity 100× less. |

## 2. Small Molecule Inhibitors of CK2 and Chemical Probe Development

### 2.1. Selected Orthosteric CK2 Inhibitors and Chemical Probes

Two chemotypes have received a lot of attention in the pursuit of orthosteric CK2 chemical probes: pyrazolo[1,5-*a*]pyrimidines (referred to as pyrazolopyrimidines from here forth) and naphthyridines. AstraZeneca was the first to publish a series of papers and patents summarizing their efforts to tune the pyrazolopyrimidine scaffold to improve CK2 potency [42–45]. Examples of lead compounds from the AstraZeneca efforts are compounds **7b** and **7h** in Table 2 and Figure 2 [43,44]. While these pyrazolopyrimidines were found to be potent inhibitors of the enzymatic activity of CK2α and later potently engage CK2α and CK2α′ in cells [17], their suboptimal kinome-wide selectivity would need to be improved to furnish a pyrazolopyrimidine-based CK2 chemical probe [17,43,44]. Next, a team at SGC Frankfurt developed a series of acyclic/macrocyclic pyrazolopyrimidines. Macrocyclization was pursued as a strategy to deliver a selective CK2 chemical probe. While the lead compound **IC20** (Figure 2 and Table 2) proved to be a potent and selective CK2 inhibitor and outperformed its acyclic counterpart **IC19** (Figure 2 and Table 2) in both of these categories, the series suffered from poor cellular penetrance that precluded the advancement of **IC20** as a CK2 chemical probe [18]. Building on the work of AstraZeneca, SGC-UNC pursued optimization of the pyrazolopyrimidine scaffold toward a CK2 chemical probe. The best available CK2 chemical probe to date was delivered from these efforts: **SGC-CK2-1** (Figure 2). This compound demonstrates potent CK2 inhibition as well as the sub-micromolar engagement of CK2 in cells (Table 2). Broad profiling revealed that **SGC-CK2-1** is exquisitely selective (Table 2): it only inhibits CK2α and CK2α′ and has a 100-fold selectivity window over the next most potently inhibited kinase (DYRK2). Screening of **SGC-CK2-1** against nearly 180 cancer cell lines challenged the essentiality of CK2 for cancer cell proliferation [17], a notion that was challenged by others in responsive publications [3,15]. Finally, **SGC-CK2-1N** (Figure 2) was released alongside **SGC-CK2-1** as a structurally related negative control compound [17].

The naphthyridine scaffold gave rise to the most clinically advanced CK2 inhibitor: **CX-4945** (Figure 2). This compound, despite its suboptimal selectivity (Table 2), has advanced into clinical trials for several oncological indications as well as for the treatment of COVID-19 [19,46–48]. Efforts have been made to improve the selectivity of naphthyridine-based CK2 inhibitors. Collaborative work between members of the University of Padova and Cylene Pharmaceuticals led to the development of **CX-5011**, **CX-5033**, and **CX-5279** (Figure 2), which retained potent biochemical inhibition of CK2 (Table 2). Selectivity profiling of these compounds versus **CX-4945** in assay panels comprising 102–235 kinases established that the more recently generated analogs are more selective than **CX-4945** [20]. Kinome-wide profiling (468 kinase panel) of **CX-4945** and **CX-5033** confirmed that these compounds do not have the selectivity exhibited by **SGC-CK2-1** (Table 2) [16,17]. Finally, when **CX-5011**, **CX-5033**, and **CX-4945** were evaluated for CK2 cellular target engagement, only **CX-4945** and **CX-5011** demonstrated sub-micromolar CK2 cellular target engagement (Table 2). Building on the structure–activity relationships developed by these efforts, the SGC-UNC team identified **SGC-CK2-2**, a naphthyridine-based CK2 chemical probe (Figure 2). Like the previously discussed naphthyridines, **SGC-CK2-2** is a potent inhibitor of CK2 in cell-free enzymatic assays (Table 2). It also demonstrates sub-micromolar CK2

cellular target engagement in the corresponding NanoBRET assays (Table 2). An important distinction from its predecessors is the kinome-wide selectivity of **SGC-CK2-2**. This compound is exquisitely selective (Table 2) and inhibits only CK2α and CK2α', with a 200-fold selectivity window over the next most potently inhibited kinase (HIPK2). Like **SGC-CK2-1**, **SGC-CK2-2** was not broadly antiproliferative when evaluated in panels of different cancer cell lines [16]. Finally, **SGC-CK2-2N** (Figure 2) was designed as a structurally related negative control compound to be used alongside **SGC-CK2-2** [16].

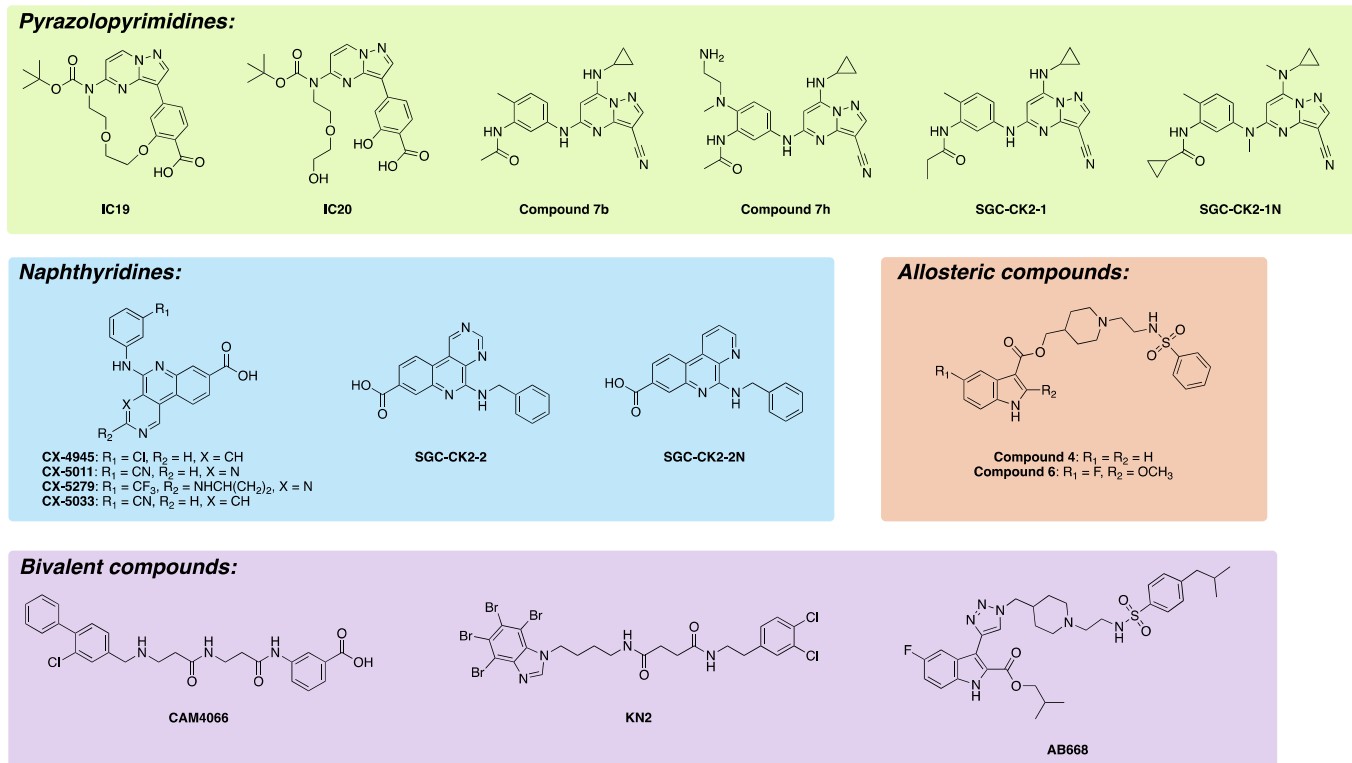

**Figure 2.** Structures of selected orthosteric, allosteric, and bivalent CK2 inhibitors discussed herein. The pyrazolopyrimidines and naphthyridines represent two chemotypes that have been extensively explored as orthosteric CK2 inhibitors.

Where possible, it is best practice to interrogate a phenotype using chemical probes built upon orthogonal chemotypes. When comparing the in vitro properties of **SGC-CK2-1** and **SGC-CK2-2** (Table 3), some advantages and disadvantages to using each in cell-based studies arise. While the two chemical probes have very similar cell-free enzymatic potency, **SGC-CK2-1** is much more potent in cells. The low nanomolar CK2 cellular target engagement IC$_{50}$ values for **SGC-CK2-1** translate to a much lower concentration needed for efficacy in cell-based studies compared with **SGC-CK2-2**. The aqueous kinetic solubility of **SGC-CK2-2**, however, is much better than that observed for **SGC-CK2-1**. Both compounds are exquisitely selective and, when used at a proper concentration, can be used to deduce that an observed phenotype is driven by the inhibition of CK2. Both CK2 chemical probes were evaluated in cell-based systems only, and thus it has been stressed that their use is limited to in vitro studies.

**Table 2.** Summary of data for selected orthosteric, allosteric, and bivalent CK2 inhibitors. Details on the binding site and the core scaffold of each inhibitor are specified, together with their biochemical and cellular affinity, if reported. Kinome-wide selectivity data are also summarized, specifying the size of the kinase panel used for selectivity screening and the results. An $S_X$ (Y μM) score indicates the proportion of the kinase panel that binds with <X percent of control remaining when dosed at a concentration of Y μM. The Gini coefficient is a score describing kinase selectivity, with perfect selectivity (inhibiting only a single target, and zero inhibition of all other targets) achieving a score of 1 and equal inhibition of all kinases achieving a score of 0. Negative controls designed to pair with identified chemical probes are also indicated, where available. One or more PDB codes for co-crystal structures with CK2 are also provided, if available. Finally, key references that describe the discovery and characterization of the inhibitor and were used to populate the table are provided as PMID or DOI identifiers.

| CK2 Inhibitor | Binding Site | Core Scaffold | Biochemical CK2 Affinity | Cellular CK2 Affinity | Selectivity | Negative Control | PDB Entries | PMIDs/DOI |
|---|---|---|---|---|---|---|---|---|
| **IC19** | Orthosteric | Pyrazolopyrimidine | CK2α ITC $K_D$ = 82 nM | CK2α NB [1] $IC_{50}$ = 1.5 μM; CK2α′ NB $IC_{50}$ = 7.4 μM | 469 kinase DiscoverX KINOMEscan $S_{35}$ (1 μM) = 0.07 | - | - | 32883634 |
| **IC20** | Orthosteric | Pyrazolopyrimidine | CK2α ITC $K_D$ = 12 nM | CK2α NB $IC_{50}$ = 1.5 μM; CK2α′ NB $IC_{50}$ = 7.6 μM | 469 kinase DiscoverX KINOMEscan $S_{35}$ (1 μM) = 0.02 | - | 6YUM | 32883634 |
| **Compound 7b** | Orthosteric | Pyrazolopyrimidine | CK2α enzymatic $IC_{50}$ < 3.0 nM | CK2α NB $IC_{50}$ = 3.2 nM; CK2α′ NB $IC_{50}$ = 1.8 nM | 402 kinase DiscoverX KINOMEscan $S_{40}$ (0.1 μM) = 0.025; 469 kinase DiscoverX KINOMEscan $S_{10}$ (1 μM) = 0.032 | - | - | 24900749, 33484635 |
| **Compound 7h** | Orthosteric | Pyrazolopyrimidine | CK2α enzymatic $IC_{50}$ < 3.0 nM; CK2α SPR $K_D$ = 6.3 pM | CK2α NB $IC_{50}$ = 5.3 nM; CK2α′ NB $IC_{50}$ = 4.4 nM | 402 kinase DiscoverX KINOMEscan $S_{50}$ (0.1 μM) = 0.03; 469 kinase DiscoverX KINOMEscan $S_{10}$ (1 μM) = 0.042 | - | 5H8E | 26985319, 33484635 |
| **SGC-CK2-1** | Orthosteric | Pyrazolopyrimidine | CK2α enzymatic $IC_{50}$ = 4.2 nM; CK2α′ enzymatic $IC_{50}$ = 2.3 nM | CK2α NB $IC_{50}$ = 36 nM; CK2α′ NB $IC_{50}$ = 16 nM | 468 kinase DiscoverX KINOMEscan $S_{10}$ (1 μM) = 0.007 | SGC-CK2-1N | 6Z83 | 33484635 |
| **CX-4945** | Orthosteric | Naphthyridine | CK2α enzymatic $IC_{50}$ = 1 nM; CK2α′ enzymatic $IC_{50}$ = 1 nM | CK2α NB $IC_{50}$ = 240 nM; CK2α′ NB $IC_{50}$ = 180 nM | 468 kinase DiscoverX KINOMEscan $S_{10}$ (1 μM) = 0.069; 235 kinases at 0.5 μM; Gini coefficient = 0.667 | - | 3NGA, 3PE1 | 33484635, 21174434, 21870818, 37077385 |

**Table 2.** *Cont.*

| CK2 Inhibitor | Binding Site | Core Scaffold | Biochemical CK2 Affinity | Cellular CK2 Affinity | Selectivity | Negative Control | PDB Entries | PMIDs/DOI |
|---|---|---|---|---|---|---|---|---|
| **CX-5011** | Orthosteric | Naphthyridine | CK2$\alpha$ enzymatic IC$_{50}$ = 2.3 nM | CK2$\alpha$ NB IC$_{50}$ = 350 nM; CK2$\alpha'$ NB IC$_{50}$ = 66 nM | 235 kinases at 0.5 $\mu$M; Gini coefficient = 0.794 | - | 3PE2 | 21870818, 37077385 |
| **CX-5033** | Orthosteric | Naphthyridine | CK2 holoenzyme enzymatic IC$_{50}$ = 4.0 nM | CK2$\alpha$ NB IC$_{50}$ = 4300 nM; CK2$\alpha'$ NB IC$_{50}$ = 1900 nM | 468 kinase DiscoverX KINOMEscan S$_{10}$ (1 $\mu$M) = 0.079; 103 kinases at 0.5 $\mu$M; Gini coefficient = 0.570 | - | - | 21174434, 21870818, 37077385 |
| **CX-5279** | Orthosteric | Naphthyridine | CK2$\alpha$ enzymatic IC$_{50}$ = 0.91 nM | - | 102 kinases at 0.5 $\mu$M; Gini coefficient = 0.755 | - | 3R0T | 21870818 |
| **SGC-CK2-2** | Orthosteric | Naphthyridine | CK2$\alpha$ enzymatic IC$_{50}$ = 3.0 nM; CK2$\alpha'$ enzymatic IC$_{50}$ < 1.0 nM | CK2$\alpha$ NB IC$_{50}$ = 920 nM; CK2$\alpha'$ NB IC$_{50}$ = 200 nM | 468 kinase DiscoverX KINOMEscan S$_{10}$ (1 $\mu$M) = 0.007 | SGC-CK2-2N | 8BGC | 37077385 |
| **Compound 4** | Allosteric | Indole-3-carboxylate | CK2$\beta$-dependent enzymatic IC$_{50}$ = 45 $\mu$M; CK2$\alpha$ K$_D$ = 41 $\mu$M | - | - | - | 6FVG | 31685885 |
| **Compound 6** | Allosteric | Indole-3-carboxylate | CK2$\beta$-dependent enzymatic IC$_{50}$ = 22 $\mu$M; CK2$\alpha$ K$_D$ = 30 $\mu$M | - | - | - | 6FVF | 31685885 |
| **CAM4066** | Bivalent | Biphenylmethylamine | CK2$\alpha$ ITC K$_D$ = 320 nM; CK2$\alpha$ enzymatic IC$_{50}$ = 370 nM | - | 52 kinases at 2 $\mu$M; Gini coefficient = 0.82 | - | 5CU3, 5CU4, 5MO8 | 28451126 |

**Table 2.** *Cont.*

| CK2 Inhibitor | Binding Site | Core Scaffold | Biochemical CK2 Affinity | Cellular CK2 Affinity | Selectivity | Negative Control | PDB Entries | PMIDs/DOI |
|---|---|---|---|---|---|---|---|---|
| **KN2** | Bivalent | Benzoimidazole | $CK2\alpha_2\beta_2$ holoenzyme enzymatic $IC_{50} = 19$ nM and $K_i = 6.1$ nM; $CK2\alpha'_2\beta_2$ holoenzyme enzymatic $IC_{50} = 16$ nM and $K_i = 4.0$ nM | - | 83 kinases at 3 μM; Gini coefficient = 0.76 | - | 7AT5 | 34323071 |
| **AB668** | Bivalent | Indole-2-carboxylate | $CK2\alpha$ binding assay $K_D = 86$ nM; $CK2\beta$-dependent enzymatic $K_i = 41$ nM | - | 468 kinase DiscoverX KINOMEscan $S_{10}$ (2 μM) = 0.007 | - | - | 10.1101/ 2022.12.16. 520736 |

[1] NB = NanoBRET.

**Table 3.** Comparison of SGC-CK2-1 and SGC-CK2-2 in terms of potency, selectivity, and solubility.

| | SGC-CK2-1 | SGC-CK2-2 |
|---|---|---|
| CK2α enzymatic IC$_{50}$ | 4.2 nM | 3.0 nM |
| CK2α′ enzymatic IC$_{50}$ | 2.3 nM | <1.0 nM |
| Selectivity: S$_{10}$ (1 μM) | 0.007 | 0.007 |
| Selectivity window | 100-fold over DYRK2 | 200-fold over HIPK2 |
| CK2α NanoBRET IC$_{50}$ | 36 nM | 920 nM |
| CK2α′ NanoBRET IC$_{50}$ | 19 nM | 200 nM |
| Kinetic solubility | 3.7 μM | 211 μM |

*2.2. Selected Allosteric and Bivalent CK2 Inhibitors in Development*

As a strategy to achieve selectivity for CK2, several groups have pursued the identification of CK2 inhibitors that bind outside of the ATP-binding site. Compounds **4** and **6** (Figure 2), which were developed from a parent compound discovered via virtual screening, are reported to bind to the CK2α/CK2β interface. SPR, crystallographic, and nMR experiments were used to confirm binding to CK2α and inhibition of CK2β-dependent enzymatic activity, albeit with mid-micromolar K$_D$/IC$_{50}$ values (Table 2). While inhibition of cancer cell growth and downstream signaling was demonstrated in response to treatment with compound **6**, CK2 cellular target engagement was not demonstrated, and a lack of selectivity data makes it difficult to interpret the results as CK2-specific [21].

**CAM4066** is a bivalent CK2 inhibitor that binds to the αD pocket on CK2α and also occupies the CK2α ATP site. This compound is an elaborated version of the original fragment hit, which was identified via a high-concentration crystallographic screen. **CAM4066** was shown to bind to CK2α via ITC with a K$_D$ = 320 nM and, correspondingly, to inhibit the enzymatic activity of CK2α with an IC$_{50}$ = 370 nM (Table 2). In a small, 52-member selectivity screen, **CAM4066** exhibited improved selectivity versus **CX-4945** (Table 2). The ability of **CAM4066** to inhibit the growth of three different cancer cell lines was examined but its cellular penetrance was questioned when no effect on cellular viability was observed [25,27]. No experiments that would provide evidence of cellular target engagement with CK2 by **CAM4066** have been provided.

Like **CAM4066**, **KN2** is a bivalent CK2 inhibitor that occupies the αD pocket and ATP site as well. **KN2**, however, was designed to bind to the CK2α′, rather than the CK2α, αD pocket and ATP site. 3,4-Dichlorophenethylamine (DPA) was used to anchor the bivalent ligand in the CK2α′ αD pocket, while a derivative of 4,5,6,7-tetrabromo-1*H*-benzo[*d*]imidazole (TBI) called TBIa engaged the CK2α′ ATP site. The two were covalently linked via succinic acid through the formation of amide bonds. **KN2** was found to bind and inhibit the CK2 holoenzymes comprising CK2α$_2$β$_2$ or CK2α′$_2$β$_2$ with low nanomolar K$_i$/IC$_{50}$ values, respectively (Table 2). The CK2 binding affinity of **KN2** was between that of **CX-4945** and **CAM4066**. Selectivity profiling of **KN2** versus 83 kinases revealed it to demonstrate a selectivity score between that of **CAM4066** (more selective) and **CX-4945** (less selective), albeit kinase panels of variable sizes and different concentrations of the inhibitor were employed in each screen (Table 2). While cell-based toxicity assays demonstrated that **KN2** is cytotoxic and, therefore, cell-permeable, CK2 cellular target engagement has not been assessed for this compound. Western blots of CK2 substrates have been used to demonstrate the inhibition of downstream CK2 signaling by **KN2** [26].

A final bivalent compound that has recently been disclosed is **AB668**. Like **CAM4066** and **KN2**, **AB668** binds both at the CK2 ATP site and the allosteric αD pocket. A thermal shift assay and X-ray crystallography were used to confirm binding to CK2α, which was then quantified using a CK2α active site-directed competition binding assay, revealing a K$_D$ = 86 nM (Table 2). **AB668** demonstrated inhibition of the CK2 holoenzyme with K$_i$ = 41 nM. The kinome-wide selectivity of **AB668** was profiled using the largest commercial panel and was found to be exquisitely selective, matching the selectivity of **SGC-CK2-1** and **SGC-CK2-2**. **AB668** was found to induce caspase-3 activation in clear cell renal cell carcinoma, elicit significant cell proliferation arrest in two cancer cell lines, and reduce

cell viability in an ex vivo culture model of renal carcinoma. Assaying the CK2 activity in treated cell extracts has been used to indirectly assess the CK2 cellular target engagement of **AB668** [28].

### 3. Use of SGC-CK2-1 by the Community in Cell-Based Studies

**SGC-CK2-1** was developed to be a useful CK2-targeting chemical probe in cells. The seminal paper on this probe did not evaluate its utility in vivo. This compound was made immediately available to all interested investigators via request from the original authors, through the SGC, and by multiple chemical vendors. Additional attention was drawn to **SGC-CK2-1** by a spotlight in *Trends in Pharmacological Sciences* [49]. Follow-up studies have been carried out with **SGC-CK2-1**, both in collaboration with the original authors and independently.

The development and validation of an assay in microglia derived from human-induced stem cells (hiPSCs) enabled side-by-side evaluation of the ability of **SGC-CK2-1** and **CX-4945** to blunt an inflammatory response. Briefly, to define the role of CK2 in microglia, wild type hiPSC-derived microglial-like cells (MGLs) and those homozygous for an autosomal dominant mutation in presenilin-1 (PSEN1ΔE9), which is causative for early-onset familial Alzheimer's disease (AD), were stimulated with lipopolysaccharides for 24 h in the absence or presence of either **SGC-CK2-1** or **CX-4945** [50]. Both mRNA and protein for IL-6 and IL1-β, two pro-inflammatory cytokines that are increased in AD patients and, when increased, correlate with cognitive impairments, were examined [50–56]. A dose–dependent and significant inhibition of the inflammatory response without associated toxicity was observed in both wild type and PSEN1ΔE9 MGLs in response to low nanomolar **SGC-CK2-1** treatment. Furthermore, **SGC-CK2-1** much more efficiently suppressed inflammatory cytokine expression than **CX-4945,** and the data suggested a difference in the effectiveness of CK2 inhibition in the wild type versus PSEN1ΔE9 MGLs [50].

**SGC-CK2-1** was also employed in a study that characterized the role of CK2 in β-coronavirus replication and entry. Alongside other pyrazolopyrimidines, the efficiency of **SGC-CK2-1** and **CX-4945** to block the replication of pathogenic human, bat, and murine β-coronaviruses was explored and confirmed. The potency of CK2α cellular target engagement by CK2 inhibitors from divergent chemical series was found to correlate with antiviral activity. Furthermore, the essential role of the CK2 holoenzyme in β-coronavirus replication was corroborated via genetic knockdown. Through the use of a spike protein endocytosis assay, **SGC-CK2-1** was also shown to suppress viral entry [4].

In a study aimed at refining the list of bona fide CK2 substrates, **SGC-CK2-1** and **CX-4945** were once again used side-by-side and compared. Using triple stable isotope labeling by amino acids in cell culture (SILAC) in combination with inhibitor-resistant CK2, a reliable method was established for the identification and validation of CK2 substrates [57]. This methodology was employed to evaluate the selectivity of **CX-4945** versus **SGC-CK2-1**. It was determined that only a minority of phosphosites that were significantly downregulated in response to **CX-4945** treatment were CSNK2A1-dependent. In comparison, the majority of phosphosites that were significantly downregulated in response to **SGC-CK2-1** treatment were CSNK2A1-dependent. The authors concluded based on their data that **SGC-CK2-1** demonstrates significantly enhanced selectivity toward CSNK2A1 compared to **CX-4945**. Through the use of inhibitor-resistant CK2 cells in tandem with **SGC-CK2-1**, more than 300 CSNK2A1-dependent phosphosites were identified [2,10].

**SGC-CK2-1** and **CX-4945** were used in parallel to evaluate the importance of CK2 in promoting insulin production and secretion from pancreatic β-cells. Without associated toxicity, both compounds downregulated the endogenous CK2 activity to a similar extent. In addition, both **SGC-CK2-1** and **CX-4945** increased the message for insulin and amplified insulin secretion from storage vesicles. The use of **SGC-CK2-1** in their experiments allowed the authors to unambiguously attribute the observed responses to inhibition of CK2 [58].

A known role of CK2 in the regulation of NG2 gene expression and the connection of NG2 expression with juvenile angiofibroma (JA) motivated the exploration of whether

CK2 inhibition could be a therapeutical avenue for JA patients. For this study, the authors explored whether CK2 inhibition suppressed NG2-dependent JA cell proliferation and migration. Via Western blot, immunohistochemistry, flow cytometry, and quantitative real-time PCR, NG2, and CK2 were found to be expressed in JA patient-derived tissue samples and JA patient-derived cells. The treatment of JA patient-derived cells with either **SGC-CK2-1** or **CX-4945** resulted in a significant reduction in the NG2 gene and protein expression, which was accompanied by reduced proliferation and migration of these cells. The use of **SGC-CK2-1** enabled the interrogation of a promising new therapeutic approach for a cohort of JA patients [59].

## 4. Development of an In Vivo CK2 Probe

### 4.1. Definition of an In Vivo Chemical Probe

While the abovementioned chemical probes are validated in vitro, it is also of interest to develop chemical probes that can be used to interrogate CK2 function in vivo. While there has not been a consensus on the properties of an in vivo chemical probe, we propose the following criteria (Table 4) in addition to the criteria required for an in vitro chemical probe (Table 1).

**Table 4.** Proposed parameters to benchmark an in vivo chemical probe.

| Proposed In Vivo Chemical Probe Criteria |
| --- |
| 1. For at least one animal model, detailed characterization is provided for pharmacokinetic parameters: bioavailability (F), half-life ($t_{1/2}$), intrinsic clearance ($CL_{int}$), the volume of distribution ($V_d$), maximum concentration ($C_{max}$) and the time taken to reach it ($t_{max}$); |
| 2. Dosing regimen optimized: dose amount, frequency of dose, route of administration, and formulation; |
| 3. Maintain a concentration $\geq 3\times$ the in vitro cellular $IC_{50}/EC_{50}$ in the plasma or relevant tissue compartment; |
| 4. Demonstrate in vivo target engagement or target inhibition in the plasma or relevant tissue compartment. |

Firstly, a detailed pharmacokinetic study would be required to investigate key pharmacokinetic parameters relevant to drug dosing in an appropriate animal model. These parameters include the bioavailability (F) of an oral (p.o.) or intraperitoneal (i.p.) route of administration. The compound concentration over time in the plasma, as well as in any relevant tissue compartments, should be characterized to determine the half-life ($t_{1/2}$), intrinsic clearance ($CL_{int}$), and maximum concentration ($C_{max}$) and the time taken to reach it ($t_{max}$). The volume of distribution ($V_d$) would also provide information about the distribution of the compound within the animal. We do not propose specific criteria or cut-offs for any of these parameters, as we recognize that these parameters are primarily relevant for the optimization of the dosing regimen, including the dose amount, frequency of dose, route of administration, and formulation. While the oral route of administration is preferred for drugs, for chemical probes intended to interrogate the effects of target inhibition in vivo, the intraperitoneal route of administration is acceptable and offers benefits to bypass absorption challenges [60]. It is crucial to note that the pharmacokinetic parameters will likely differ by animal species and strain. However, given the diversity of animal models used for different studies, it would be cost- and time-prohibitive to characterize such parameters and optimize dosing regimens across multiple animal models. Therefore, we emphasize that the prudent use of in vivo chemical probes across different animal models would require measuring the pharmacokinetic parameters and adjusting the dosing regimen in each case. However, the data provided in one animal model would ideally provide a clear starting point to enable optimization of the dosing regimen for different models. For this reason, we recommend detailed characterization and reporting of pharmacokinetic parameters as a criterion for an in vivo chemical probe (Table 4).

An optimal dosing regimen should maintain the compound concentration at least 3-fold higher than the cellular $IC_{50}/EC_{50}$ in the plasma or the relevant tissue compartment. The duration of which the necessary compound concentration is maintained should be taken into account when optimizing dosing frequency. Finally, for an in vivo chemical probe, it is also important to demonstrate either target engagement or target inhibition in plasma or a relevant tissue compartment and relate this with concentration measured for a pharmacokinetic–pharmacodynamic (PK–PD) analysis [61]. This could be performed by measuring a biochemical effect of inhibiting the target (e.g., downstream phosphorylation of a phosphosite specific to the kinase inhibited), or through other target engagement techniques, such as the cellular thermal shift assay (CETSA) [62,63]. We summarize a general workflow that aligns with our above proposal to arrive at an in vivo chemical probe (Figure 3).

### 4.2. Progress toward an In Vivo Chemical Probe

Efforts have been made to develop an in vivo chemical probe using the pyrazolopyrimidine scaffold. AstraZeneca reported the inhibition of in vivo downstream phosphorylation of AKT$^{S129}$ by compounds **7b** and **7h** in mouse xenografts [43,44]. These compounds, however, demonstrate suboptimal selectivity, and their pharmacokinetic parameters are not reported, precluding characterization of their suitability as in vivo chemical probes. As mentioned previously, **SGC-CK2-1** is a great cellular tool but is recognized only as an in vitro chemical probe. Efforts to optimize **SGC-CK2-1** for in vivo use were hindered by its moderate aqueous solubility and rapid metabolism, where 60% of the compound was degraded in vitro in mouse liver microsomes with 30 min of incubation [64]. Even after successful medicinal chemistry optimization to reduce metabolism in mouse liver microsomes, the SGC-UNC team found that Phase II metabolism in mouse hepatocytes in vitro and in mice (in vivo) contributed significantly to the rapid metabolism of compounds of the pyrazolopyrimidine scaffold [64]. While it was possible to improve the exposure of pyrazolopyrimidines in vivo by co-dosing with CYP450 inhibitor 1-aminobenzotriazole and GST inhibitor ethacrynic acid, co-dosing with these broad metabolic inhibitors would likely affect other physiological processes unrelated to CK2 inhibition, and is thus unsuitable for the study of CK2 in vivo [64]. Further work is still ongoing to optimize the pyrazolopyrimidines for use in vivo without the need to co-dose with metabolic inhibitors.

Relative to the pyrazolopyrimidines, the naphthyridine series has been used in vivo more frequently. Of the naphthyridines, **CX-4945** is the frontrunner as a drug candidate under clinical investigation [47,65]. **CX-4945** demonstrated an excellent pharmacokinetic profile, with 20–51% oral bioavailability and long half-lives of 5.0–11.6 h in mice, rats, and dogs. This outstanding pharmacokinetic profile also extends to other naphthyridines, such as **CX-5011**, which demonstrates 44% oral bioavailability and 3.3–6.4 h half-life in rats and dogs [66]. When dosed in vivo, **CX-4945** also demonstrated tumor growth inhibition in mouse xenograft models with oral dosing [19,67], concurrent with a decrease in phosphorylation of p21$^{T145}$ in the tumor xenografts [67,68]. **CX-4945** also demonstrated permeability through the blood–brain barrier, which was reflected by the inhibition of CK2 substrates in intracranial xenograft models [6]. This is especially exciting as CK2 is a key player in neurodevelopment and in multiple neurological disorders [8,69], including Alzheimer's disease [5,7,70–73], and having a potent and selective, brain-penetrant in vivo tool molecule for CK2 inhibition would be extremely valuable. The limited selectivity of **CX-4945** must be acknowledged, and its application as an in vivo tool to attribute phenotypic effects to CK2 inhibition must be considered with caution. For example, DYRK1A is one of the potent off-targets of **CX-4945** [17,74]. Consequently, the in vivo inhibition of Tau$^{T212}$, a substrate of DYRK1A, was confirmed in mice, and **CX-4945** rescued neurological defects from overexpression of *mnb*, the ortholog of DYRK1A, in *Drosophila* [74]. It is important to note that the strain of mice and the doses of **CX-4945** used in this study were similar to those for the aforementioned in vivo studies. As such, both the inhibition of CK2 in this study and the inhibition of DYRK1A in the other in vivo studies are expected, and any phenotypic effect

cannot be solely attributed to the inhibition of either CK2 or DYRK1A. This accentuates the need for selectivity in the criteria for a chemical probe and highlights the importance of utilizing orthogonal chemical probes when possible. As the naphthyridine compounds have better pharmacokinetic properties than the pyrazolopyrimidines, optimization based on a selective starting point, such as **SGC-CK2-2,** is a promising path forward toward an in vivo chemical probe for this chemical series.

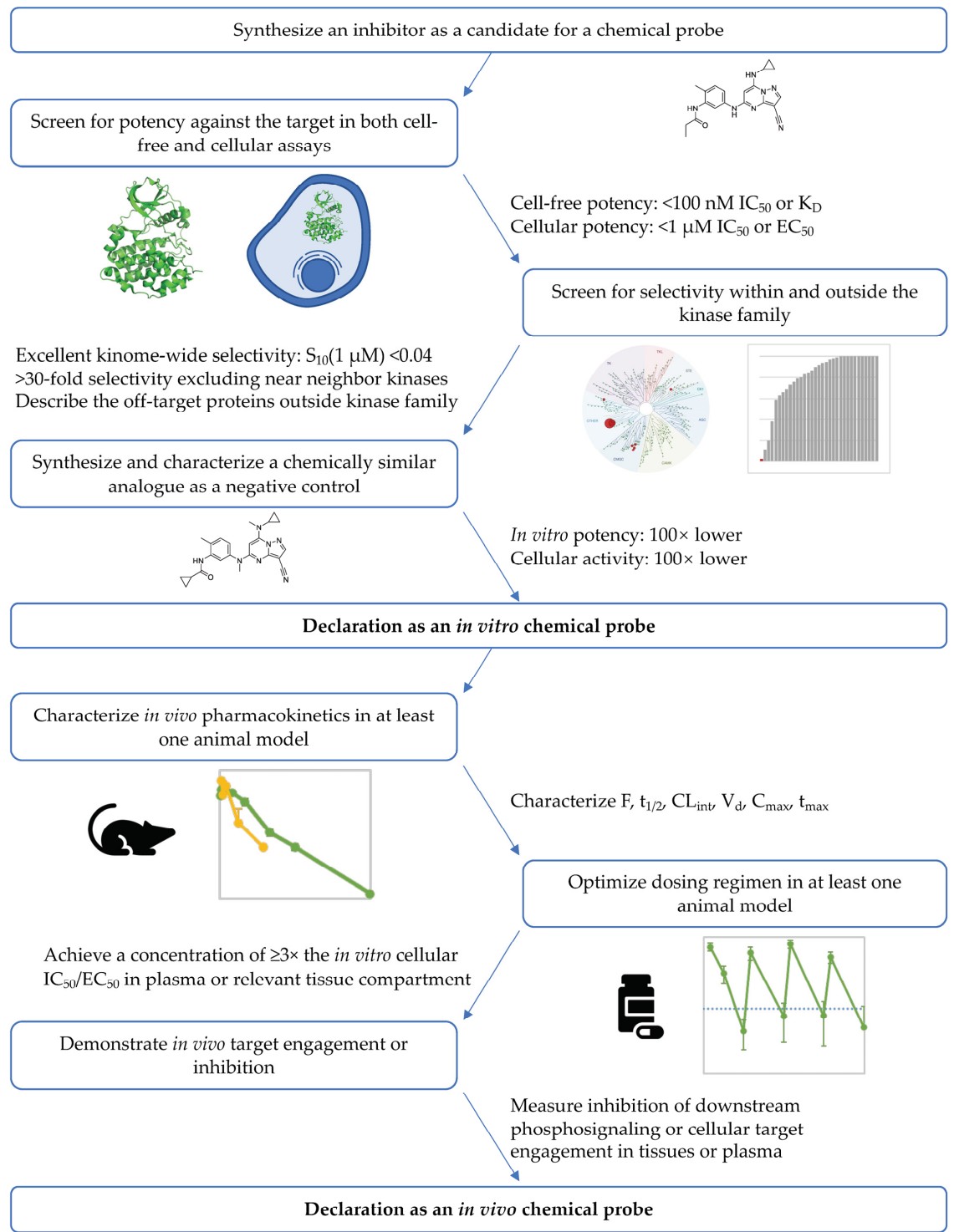

**Figure 3.** A proposed workflow for the development of an in vivo chemical probe, starting from a candidate compound. Characterization as an in vitro chemical probe and then in vivo characterization as an in vivo chemical probe are the next steps in the process.

Another CK2 inhibitor under clinical investigation is **CIGB-300** (also known as **CIGB-325**), a cell-permeable peptide that binds to CK2 substrates [29,75] and to CK2 directly [76,77] to prevent CK2-induced phosphorylation. **CIGB-300** demonstrated the inhibition of tumor growth in multiple cancer cell lines in vitro and also in vivo when administered via intratumor injections [29]. A follow-up study showed that **CIGB-300** may also be administered via intraperitoneal or intravenous routes and accumulates in tumors [78]. **CIGB-300** demonstrated preliminary efficacy in clinical investigations [79–82], lending support for its tolerability and efficacy in vivo. Because of the unique mechanism of action of **CIGB-300**, it offers an orthogonal approach to interrogate CK2 activity in vivo. If efforts are made to fully characterize the binding partners and mechanism of action of **CIGB-300**, it may demonstrate utility as a chemical probe.

## 5. Conclusions

The two clinically investigated inhibitors, **CX-4945** and **CIGB-300,** are the culmination of the extensive research invested in the development of CK2 inhibitors as drugs with a focus on oncology. The inhibition of tumor growth in response to CK2 inhibitors is often attributed to CK2 inhibition. Yet, full characterization of the kinome-wide inhibitory profile of traditional CK2 inhibitors reveals that some, including **CX-4945**, are not sufficiently selective to allow attribution of a phenotype to CK2. A more selective chemical probe is thus required for investigating CK2 function. The development of **SGC-CK2-1** and **SGC-CK2-2**, two highly selective CK2 chemical probes from distinct chemotypes, addresses this need and helps to deconvolute responses driven by CK2 inhibition. In particular, the lack of broad-spectrum antitumor activity of **SGC-CK2-1** calls into question if the antitumor effects of **CX-4945** are indeed caused by the on-target inhibition of CK2 alone [17]. The adoption of **SGC-CK2-1** as a CK2 chemical probe by the scientific community for use in cell-based studies is a positive step forward. We emphasize that the best practice of using a chemical probe is in parallel with its associated negative control, and/or the use of two orthogonal chemical probes [83]. Both options are currently enabled for CK2.

CK2 chemical probes can still be further improved. Of the allosteric and bivalent inhibitors reported, **AB668** demonstrates the best selectivity profile. This represents a promising direction toward the development of a third CK2 chemical probe, with an orthogonal scaffold to the two currently established chemical probes and differential binding mode. The peptidic inhibitor **CIGB-300** is also a potential candidate orthogonal chemical probe for CK2, but further characterization of its binding partners is required due to its unique mechanism of action. It is also of interest to develop probes for in vivo study of CK2 function, and much work still needs to be performed in this area. While optimization of pyrazolopyrimidines was met with challenges due to metabolic instability, the naphthyridines have demonstrated an improved pharmacokinetic profile. Further optimization of their in vivo potency would potentially lead to a suitable chemical probe from the naphthyridine series for use in living organisms.

The use of CK2 chemical probes has shed much light on the function of this pleiotropic kinase. Two chemical probes have been vetted and embraced so far, but the development of additional chemical probes would allow the scientific community to perform better experiments, in vitro or in vivo. These basic science experiments with high-quality probes will enable the community to determine which diseases are candidates for treatment with selective CK2 inhibitors.

**Author Contributions:** Conceptualization, D.H.D. and A.D.A.; methodology, H.W.O. and D.H.D.; investigation, H.W.O. and A.D.A.; writing—original draft preparation, H.W.O. and A.D.A.; writing—review and editing, all authors.; visualization, H.W.O. and A.D.A.; supervision, D.H.D. and A.D.A.; project administration, D.H.D. and A.D.A.; funding acquisition, D.H.D. and A.D.A. All authors have read and agreed to the published version of the manuscript.

**Funding:** The Structural Genomics Consortium (SGC) is a registered charity (number 1097737) that receives funds from Bayer AG, Boehringer Ingelheim, the Canada Foundation for Innovation, the Eshelman Institute for Innovation, Genentech, Genome Canada through the Ontario Genomics Institute, EU/EFPIA/OICR/McGill/KTH/Diamond, Innovative Medicines Initiative 2 Joint Undertaking, Janssen, Merck KGaA (aka EMD in Canada and USA), Pfizer, the São Paulo Research Foundation-FAPESP, and Takeda. Research reported in this publication was supported in part by NIH U24DK116204 and DoD ALSRP award AL190107. The North Carolina Policy Collaboratory at the University of North Carolina at Chapel Hill with funding from the North Carolina Coronavirus Relief Fund established and appropriated by the North Carolina General Assembly and a grant from Takeda contributed to work reviewed herein.

**Institutional Review Board Statement:** Not applicable.

**Informed Consent Statement:** Not applicable.

**Data Availability Statement:** Not applicable.

**Conflicts of Interest:** The authors declare no conflict of interest.

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
