# Peer review of "CK2 Chemical Probes: Past, Present, and Future"

_2813-3757, doi:10.3390/kinasesphosphatases1040017_

Round 1
Reviewer 1 Report
Comments and Suggestions for Authors
Here the authors reviewed small molecule inhibitors reported for protein kinase CK2 (CK2) as potential drug candidates and/or chemical probes for CK2 research, focusing their attention on pyrazolopyrimidines and naphthyridines. They also propose criteria for an ideal in vivo CK2 chemical probe and discuss future study directions. This manuscript thus contains useful information for the development of CK2 chemical probes.
There are no major comments on this manuscript.
Minor comments are as follows:
1) p.1, lines 9 and 26:
The authors use the enzyme names “Casein kinase 2” (p. 1, line 9) and “Protein kinase casein kinase 2” (p. 1, line 26). Because the name “casein kinase 2” was renamed “protein kinase CK2” by the IUBMB Nomenclature Committee, it is likely that the use of the enzyme name “protein kinase CK2” is preferable.
2) Several abbreviations [e.g., AAK1 (p. 4, line 84), BIKE (p. 4, line 84), and SILAC (p. 9, line 249)] are used without their definition. Their definitions seem to be necessary.
Comments on the Quality of English Language
During the review, the following points were noticed; please confirm them.
1) p. 2, line 48:
“CK2 ATP” seems to be “CK2 ATP site”.
2) p.6, line 150:
The description “only inhibits only CK2a …” seems to be “only inhibits only CK2a …”
3) p. 6, line 167:
“select” seems to be “selected”.
4) p. 11, line 311:
The description “The duration of which this is maintained should be …” seems to be “The duration during which the compound concentration is maintained should be …”.
Author Response
We thank the reviewer for the insightful comments and have addressed all of them as follows.
Minor comments are as follows:
1) p.1, lines 9 and 26:
The authors use the enzyme names “Casein kinase 2” (p. 1, line 9) and “Protein kinase casein kinase 2” (p. 1, line 26). Because the name “casein kinase 2” was renamed “protein kinase CK2” by the IUBMB Nomenclature Committee, it is likely that the use of the enzyme name “protein kinase CK2” is preferable.
- We have added “protein kinase” to p. 1, line 9 and then continue to use the abbreviation CK2 throughout the manuscript. For completeness’ sake, we spelled out the full name of CK2 at the start of the abstract and the introduction. These names have been used interchangeably throughout the literature and we believe that it should be acceptable.
2) Several abbreviations [e.g., AAK1 (p. 4, line 84), BIKE (p. 4, line 84), and SILAC (p. 9, line 249)] are used without their definition. Their definitions seem to be necessary.
- We have spelled out the full names of these abbreviations in our revised manuscript.
During the review, the following points were noticed; please confirm them.
1) p. 2, line 48:
“CK2 ATP” seems to be “CK2 ATP site”.
- We have changed this sentence to read “CK2 ATP-binding pocket and αD pocket”.
2) p.6, line 150:
The description “only inhibits only CK2a …” seems to be “only inhibits only CK2a …”
- We have corrected this typographical error.
3) p. 6, line 167:
“select” seems to be “selected”.
- We have changed “select” to “selected” in the Table 2 legend.
4) p. 11, line 311:
The description “The duration of which this is maintained should be …” seems to be “The duration during which the compound concentration is maintained should be …”.
We have reworded the sentence and it now reads: “The duration of which the necessary compound concentration is maintained should be…”
Reviewer 2 Report
Comments and Suggestions for Authors
The authors review the small molecule inhibitors reported for CK2, discussing various orthosteric, allosteric, and bivalent inhibitors of CK2. They focuses on the pyrazolo[1,5-a]pyrimidines and naphthyridines, two chemotypes that have been extensively explored for chemical probe development. They highlight the uptake and demonstrated utility of the pyrazolo[1,5-a]pyrimidine chemical probe SGC-CK2-1 by the scientific community in cellular studies. Albeit, I consider these findings to provide new insight into chemical-related fields, I still have some suggestions.
1, Most figures and tables are highly professional; however, the authors should guide the readers to the meaning of the images appropriately; otherwise, it will likely cause misunderstandings. Therefore, I suggest the author consider revising these figures and table legends again.
2, The author proposes criteria for an ideal in vivo chemical probe for investigating CK2 function in a living organism. They discuss ongoing and future directions in the development of in vivo chemical probes for CK2. However, it would be much better if the authors could provide some Workflow or Scheme for this research, I suggest that they take a look at the recent paper in MDPI (PMID: 35563422, 36677020, 34834441)
3, There are few typo issues for the authors to pay attention to; please also unify the writing of scientific terms. “Italic, capital”? Please double-check superscripts and subscripts for the whole manuscript.
Author Response
We thank the reviewer for the helpful comments, which have improved our manuscript, and have addressed them as follows.
1, Most figures and tables are highly professional; however, the authors should guide the readers to the meaning of the images appropriately; otherwise, it will likely cause misunderstandings. Therefore, I suggest the author consider revising these figures and table legends again.
- We have elaborated and added more detailed descriptions into the figure and table legends where it was lacking.
2, The author proposes criteria for an ideal in vivo chemical probe for investigating CK2 function in a living organism. They discuss ongoing and future directions in the development of in vivo chemical probes for CK2. However, it would be much better if the authors could provide some Workflow or Scheme for this research, I suggest that they take a look at the recent paper in MDPI (PMID: 35563422, 36677020, 34834441)
- We have added a proposed workflow (Figure 3) for the development of in vivo chemical probes to the manuscript.
3, There are few typo issues for the authors to pay attention to; please also unify the writing of scientific terms. “Italic, capital”? Please double-check superscripts and subscripts for the whole manuscript.
- We have corrected the typos, standardized italicized text for “in vivo” and “in vitro”, capitalization for “X-ray”, and checked for superscripts and subscripts throughout the manuscript.